# Electrocatalytic Properties of Mixed-Oxide-Containing Composite-Supported Platinum for Polymer Electrolyte Membrane (PEM) Fuel Cells

**DOI:** 10.3390/ma15103671

**Published:** 2022-05-20

**Authors:** Ilgar Ayyubov, Emília Tálas, Khirdakhanim Salmanzade, Andrei Kuncser, Zoltán Pászti, Ștefan Neațu, Anca G. Mirea, Mihaela Florea, András Tompos, Irina Borbáth

**Affiliations:** 1Institute of Materials and Environmental Chemistry, Research Centre for Natural Sciences, Eötvös Loránd Research Network (ELKH), Magyar Tudósok körútja 2, H-1117 Budapest, Hungary; ayyubovi@gmail.com (I.A.); talas.emilia@ttk.hu (E.T.); ksalmanzade5@gmail.com (K.S.); paszti.zoltan@ttk.hu (Z.P.); borbath.irina@ttk.hu (I.B.); 2Department of Physical Chemistry and Materials Science, Faculty of Chemical Technology and Biotechnology, Budapest University of Technology and Economics, Műegyetem rkp. 3., H-1111 Budapest, Hungary; 3Department of Inorganic and Analytical Chemistry, Faculty of Chemical Technology and Biotechnology, Budapest University of Technology and Economics, Műegyetem rkp. 3., H-1111 Budapest, Hungary; 4National Institute of Materials Physics, 405A Atomistilor Street, 077125 Magurele, Romania; andrei.kuncser@infim.ro (A.K.); stefan.neatu@infim.ro (Ș.N.); anca.coman@infim.ro (A.G.M.); mihaela.florea@chimie.unibuc.ro (M.F.)

**Keywords:** composite support, sol-gel synthesis route, TiMoO_x_, Pt electrocatalysts, hydrogen oxidation reaction, oxygen reduction reaction, Tafel slope, reaction mechanism, stability

## Abstract

TiO_2_-based mixed oxide–carbon composite supports have been suggested to provide enhanced stability for platinum (Pt) electrocatalysts in polymer electrolyte membrane (PEM) fuel cells. The addition of molybdenum (Mo) to the mixed oxide is known to increase the CO tolerance of the electrocatalyst. In this work Pt catalysts, supported on Ti_1−x_Mo_x_O_2_–C composites with a 25/75 oxide/carbon mass ratio and prepared from different carbon materials (C: Vulcan XC-72, unmodified and functionalized Black Pearls 2000), were compared in the hydrogen oxidation reaction (HOR) and in the oxygen reduction reaction (ORR) with a commercial Pt/C reference catalyst in order to assess the influence of the support on the electrocatalytic behavior. Our aim was to perform electrochemical studies in preparation for fuel cell tests. The ORR kinetic parameters from the Koutecky–Levich plot suggested a four-electron transfer per oxygen molecule, resulting in H_2_O. The similarity between the Tafel slopes suggested the same reaction mechanism for electrocatalysts supported by these composites. The HOR activity of the composite-supported electrocatalysts was independent of the type of carbonaceous material. A noticeable difference in the stability of the catalysts appeared only after 5000 polarization cycles; the Black Pearl-containing sample showed the highest stability.

## 1. Introduction

Clean energy technologies can provide a contribution for a sustainable future while bringing new technical solutions to life. Well-known examples are the rechargeable batteries and fuel cells as electrochemical generators, which are often complementary in terms of usability. Redox flow batteries (RFB) are regarded as promising electrochemical energy storage devices for grid-scale systems [1]. Battery technologies are widely used for various light-duty power sources such as mobile phones, laptops etc., as well as vehicles for short distances. In contrast, fuel cells using energy-dense fuels (e.g., hydrogen or methanol) have a clear advantage of long-distance travel or heavy-duty operation ([2] and the references cited therein). Polymer electrolyte membrane (PEM) fuel cells are suitable for vehicle applications because they have a high power density, operate with high efficiency under different load conditions including frequent start up and shut down, their working temperature is relatively low when compared to other fuel cell types and they start up quickly [3]. In a recent study, the usability of PEM fuel cells and Li-ion battery for unmanned aerial vehicles was analyzed [4]. The PEM fuel cell was found to be advantageous in that it could provide power directly without a complex system and the required specific power appeared to be attainable. Use of a Li-air battery pack would have the longest flight time and simplest configuration [4]. Hybrid electric vehicles (lithium batteries—fuel cell) may open interesting possibilities in various fields of transport such as minibus [5] locomotive [6], and small-size passenger ferry [7] applications. The integration of hydrogen fuel cells and Li-ion batteries with photovoltaic applications [8] or mobile robots [9] has also been mentioned.

The involvement of electrocatalysts is essential for the processes taking place both on the anode and cathode sides of the PEM fuel cell. When compared to hydrogen oxidation reaction (HOR) at the anode side, the oxygen reduction reaction (ORR) of the cathode catalysts is a sluggish process. In general, the ORR on the cathode electrocatalyst can proceed either via a two-proton–electron pathway resulting in H_2_O_2_, or via a four-proton–electron pathway, reducing oxygen to water [10]. The four-electron pathway can proceed via direct and indirect mechanisms [11,12,13]; the indirect routes involve H_2_O_2_ formation as an intermediate product. The direct four-electron reduction is regarded as the preferred ORR mechanism because it completely avoids H_2_O_2_ production, which can form harmful species, and it enhances the energy conversion efficiency in the fuel cell device [12,14]. Estimations taking into account these mechanistic considerations indicate that Pt is the most suitable metal for ORR catalysts among pure metals [11,12]. However, Pt has a stronger interaction with oxygenated species than the optimal binding energy for the highest ORR activity. Since the electronic structure of the Pt surface can be adjusted by alloying Pt with another metal, Pt alloy systems have become a new milestone and have been successfully applied in PEM fuel cells ([15] and the references cited therein). Polyhedron-engineered Pt-alloy nanocrystals have been demonstrated to have unprecedented electrocatalytic activity [16].

Opinions are divided on whether the availability of Pt is a bottleneck for the automotive industry and in clean energy technologies [3,17], but it is clear that the widespread use of PEM fuel cells is currently strongly influenced by its price. It has been calculated that electrodes account for nearly half of the cost [2,18,19], while the catalyst itself accounts for about 40% of the total cost [17]. Extensive research is underway worldwide to develop less expensive PEM fuel cells and/or increase the lifetime of PEM fuel cells, an important area of which is catalyst development [17].

One direction is the study of non-platinum-based catalyst electrode materials [18], the other is to reduce the amount of platinum by increasing its utilization [20]. Many studies are aimed at investigating the degradation of the most widely used Pt/C catalysts regarding both metal nanoparticles [21,22,23] and carbon supports [21,24]. Furthermore, Pt/C recycling process has been developed [25,26].

The activity loss of Pt/C catalysts was connected to the corrosion of carbon and the subsequent sintering and dissolution of Pt nanoparticles in acidic media [27,28]. The consequence of both processes is the loss of the active surface of the Pt. A possible solution for the preparation of supported Pt catalysts with increased stability i.e., increased life time is the introduction of novel supports. In the last decade, a range of oxide-containing electrocatalyst supports were proposed for both the HOR and the ORR [29]. Because of the strong metal–support interaction (SMSI), the metal oxides are capable of stabilizing the active metal in a highly dispersed state, and might help to suppress Pt dissolution. In addition, oxide substrates are not prone to oxidation and degradation, as are carbon supports. However, even if in electrochemical experiments many of them showed excellent properties, their utilization in PEM fuel cells remains extremely rare.

TiO_2_-containing materials have received special attention in this approach [30], although the insulating nature of TiO_2_ and the typically low surface area of the oxide supports still cause issues. The doping of TiO_2_ with certain transition metals was found to improve the electrical conductivity of the support while providing valuable co-catalytic function such as CO tolerance. Tungsten [31,32], molybdenum [33,34], niobium [35,36], tantalum [37,38], and tin [39] were identified as being particularly promising dopants.

A possible solution for increasing the surface area and the conductivity of the support while preserving the stabilization function of the TiO_2_ is achieved by composite formation with carbon. For example, it was demonstrated [40] that the titania layer of the Pt/TiO_2_@carbon nanotube (CNT) and Pt/TiO_2_ catalysts stabilized Pt particles much better against growth/agglomeration during accelerated degradation tests when compared to Pt/C and Pt/CNT catalytic systems. After stability tests, used for simulation of the degradation under start–stop conditions, the percentage of Pt particle size increase due to particle growth and agglomeration increased in the following order: Pt/TiO_2_ (12%) < Pt/TiO_2_@CNT (24%) < Pt/C (36%) < Pt/CNT (44%). It may be interesting to note here that utilization of carbon–metal oxide composite electrodes seems to be beneficial in other types of fuel cells working at low temperatures as well, such as in microbial fuel cells [41].

A further step towards multifunctional, corrosion-resistant and stable Pt electrocatalysts is the idea of non-noble metal-doped TiO_2_—active carbon composite supports developed in our previous research. In this approach, TiO_2_ is responsible for stabilization of the Pt nanoparticles, the doping metals (e.g., W, Mo) provide co-catalytic function, while the active carbon ensures the good conductivity and the large surface area of the electrocatalyst. In our previous works [42,43], we found that the exclusive incorporation of the doping metals into substitutional sites of the TiO_2_ lattice is the key for the enhanced stability of the mixed oxide-carbon composite-supported electrocatalysts, as it protects them from dissolution while they can still provide CO tolerance [44,45,46,47,48,49,50]. In a single cell test experiment using hydrogen contaminated with 100 ppm CO, the performance of the Pt/Ti_0.7_M_0.3_O_2_-C (M = W, Mo) catalysts was better than that of the reference Pt/C or PtRu/C catalysts [49].

In reference [50], an optimized route for the preparation of novel TiO_2_-rutile-based Ti_0.8_Mo_0.2_O_2_-C multifunctional composite support materials with different mixed oxide/carbon ratios (Ti_0.8_Mo_0.2_O_2_/C = 75/25, 50/50 and 25/75) was elaborated using Vulcan XC-72 (V), unmodified Black Pearls 2000 (BP), and functionalized BP (F-BP) carbon materials. As demonstrated by X-ray diffraction, by use of the optimized synthesis route, almost complete Mo incorporation was achieved, in spite of the widely differing structural and surface chemical characteristics of the carbon materials. The electrochemical results revealed performance differences between the electrocatalysts with different mixed oxide/carbon ratio while confirming that the catalytic properties of the system are mainly determined by the Pt–Mo interactions. Thus, an increase of the mixed oxide content in the composites to 50 and 75 wt.%, leading to a pronounced enhancement of Pt-Mo interactions, resulted in better tolerance of the catalysts to CO as compared to those with high carbon content. On the other hand, the enhanced long-term stability of the catalysts with high (75 wt.%) carbon content was attributed to their more homogeneous microstructure. Considering also the fact that a high oxide content in the catalyst layer can lead to a slight increase of the cell resistance, the BP- and F-BP-based Pt electrocatalysts with Ti_0.8_Mo_0.2_O_2_/C = 25/75 ratio seemed to be more promising for general use.

After establishing the synthesis route for the composite supports with varying dopants, varying oxide/carbon ratios, and varying types of carbon backbones, the next logical step towards the implementation of the catalysts prepared on the novel supports is their assessment in fuel cell test devices. However, testing in fuel cells requires relatively high amounts of catalysts, which makes it necessary to scale up the synthesis procedure. One of the purposes of the present work was to verify the capability of our synthesis procedure for scaling-up. Accordingly, in this work the most promising catalysts in terms of their long-term stability, namely Pt/Ti_0.8_Mo_0.2_O_2_-C systems on composites with 25 wt.% oxide and 75 wt.% carbon content with different types of carbon backbones (see references [47,50]), were prepared in an elevated (1 g) quantity and their physicochemical properties were compared to those described in our previous works. More importantly, the activity of these catalysts was investigated both in HOR and ORR by the rotating disc electrode method in order to identify the most promising system for further scaling-up and testing under the working conditions of portable PEM fuel cells.

## 2. Materials and Methods

### 2.1. Materials

Titanium–isopropoxide (Ti(O-*i-Pr*)_4_, Sigma-Aldrich, St. Louis, MO, USA, 97%), ammonium heptamolybdate tetrahydrate ((NH_4_)_6_Mo_7_O_24_ × 4H_2_O, Merck Darmstadt, Germany, 99%), and hexachloroplatinic acid hexahydrate (H_2_PtCl_6_ × 6H_2_O, Sigma-Aldrich, St. Louis, MO, USA, 37.5% Pt) were used as Ti, Mo, and Pt precursor compounds, respectively. A 5% Nafion^®^ dispersion (DuPont™ Nafion^®^ PFSA Polymer Dispersions DE 520, The Chemours Company, Wilmington, DE, USA) was used for catalyst ink preparation. Other chemicals included nitric acid (HNO_3_, 65%, a.r.), ethanol (99.55%), 2-propanol (*i*-C_3_H_5_OH, 99.9 V/V%, a.r), ethylene–glycol (EG, 99.8%), sodium borohydride (NaBH_4_, 99.95%) (all from Molar Chemicals, Halásztelek, Hungary). Glucose (Reanal, Budapest, Hungary, a lt.), sodium hydroxide (NaOH, Reanal, Budapest, Hungary > 98%), and sulfuric acid (H_2_SO_4_, Merck, Darmstadt, Germany, 96% p.a). BP and V carbons (both from Cabot, Boston, MA, USA) were used as starting carbonaceous materials. F-BP carbon was prepared as we described before [47]. Briefly, BP previously pre-treated in nitrogen (5.0 purity, Linde Gáz Magyarország Zrt, Répcelak, Hungary) at 1000 °C was modified using a two-step treatment with HNO_3_ and glucose. Commercial 20 wt.% Pt on V support (Pt/C, C-20-Pt from QuinTech, Göppingen, Germany) was used as a reference electrocatalyst.

### 2.2. Preparation of Composite Type Supports and Electrocatalysts

In our previous studies [47,50], an optimized route for preparation of Ti_0.8_Mo_0.2_O_2_-C composite supports with 25/75 mixed oxide/carbon mass ratio using various carbonaceous materials was elaborated. The multistep sol-gel-based synthesis method of the aforementioned novel electrocatalyst supports includes three main steps: (i) the low temperature deposition of TiO_2_-rutile nuclei on the carbon backbone completed by an aging step, (ii) introduction of the Mo precursor, and (iii) incorporation of the Mo into the TiO_2_-rutile crystallites using a high-temperature treatment step (HTT: Ar, 600 °C, 8 h) (see references [43,47,50] and Appendix A).

Based on our previous work [50], three promising Pt/Ti_0.8_Mo_0.2_O_2_-C (C:V, BP, and F-BP) catalysts were selected for detailed electrochemical study (see Table 1). Throughout this paper the 25 wt.% Ti_0.8_Mo_0.2_O_2_-75 wt.% C composites are designated by a unique identifier consisting of a number indicating the nominal weight percentage of the carbon with respect to the mixed oxide content, along with the type of carbon used. Accordingly, e.g., 75V is a composite of 25 wt.% mixed oxide and 75 wt.% V carbon (see Table 1). In all cases, the desired Ti/Mo atomic ratio was 80/20.

The developed preparation method makes it possible to obtain about 1 g of composite material in one batch. To ensure that the synthesis was successful, prior to Pt deposition, the Ti_0.8_Mo_0.2_O_2_-C composites were characterized by powder X-ray diffraction (XRD) and nitrogen adsorption measurements. The results of the physicochemical characterization of the unified materials are summarized in Table 1. The comparison of these results with our previous ones presented in reference [50] shows a good reproducibility of the synthesis.

Composite support materials were loaded with 20 wt.% Pt via a modified NaBH_4_-assisted ethylene–glycol (EG) reduction–precipitation method (for details see ref. [43] and Appendix A), which has been proven to result in highly dispersed Pt nanoparticles with average particle size of ~2 nm on the surface of our composites even at a relatively high Pt load [43]. The use of a reduction temperature of 65 °C permitted us to decrease the overall reaction time [51].

The Pt-loading method gave the best results (in term of Pt dispersion and uniformity) with 0.2 g of composite support material in one run. An appropriate amount of electrocatalysts sufficient for detailed characterization by various techniques was obtained in four—five batches. The characterization of different batches of Pt electrocatalysts by cyclic voltammetry (CV) was done before unifying them (see Appendix A, which confirms the good reproducibility of the Pt loading obtained for different batches). As shown in Table 1, our reduction–deposition method applied for Pt deposition results in highly dispersed nanoparticles with an average Pt size of about 2–3 nm.

### 2.3. Physicochemical Characterization of the Composite Supports and the Electrocatalysts

The details of the physicochemical characterization of the composite supports and the electrocatalysts was completed by X-ray powder diffraction (XRD), nitrogen physisorption measurements, and inductively coupled plasma–optical emission spectrometry (ICP-OES) techniques, which were performed using the same equipment and methods as in our previous study [50], and are presented in the Appendix A.

Transmission electron microscopy (TEM) studies of the samples were made by use of a JEOL 3010 high resolution transmission electron microscope (Tokyo, Japan) operating at 300 kV (see Appendix A for details).

Scanning electron micrographs of the samples were recorded with a scanning electron microscope Vega II LMU model from Tescan (Brno, Czech Republic), equipped with an energy dispersive X-ray spectrometer (EDX) Bruker Quantax 200 (Bruker Physik-AG, Karlsruhe, Germany) (see Appendix A for details).

All Raman spectra were recorded on a LabRAM HR Evolution spectrometer from Horiba Jobin Yvon (HORIBA France SAS), with a laser radiation at wavelength of 633 nm (see Appendix A for details).

X-ray photoelectron spectroscopy (XPS) investigations were carried out using the same equipment as in our previous study [50] (see details in the Appendix A). Spectra were processed with the CasaXPS [52] and the XPSMultiQuant [53] packages.

### 2.4. Electrochemical Characterization of Composite-Supported Electrocatalysts

The electrochemical characterization in a conventional three-electrode electrochemical glass cell was carried out using the same equipment and methods as in our previous studies [47,50] (see Appendix A for details). The reference electrode was a hydrogen electrode that was immersed in the same electrolyte as the working electrode. The working electrode was, prepared by supporting the electrocatalysts on glassy carbon; Pt was used as counter electrode. All potentials are given on the reversible hydrogen electrode (RHE) scale. The electrolyte was 0.5 M H_2_SO_4_. The Pt loading of the electrodes was 10 µg cm^−2^. The details of the preparation of the working electrode, the composition of the catalyst ink, the activation procedure, and the electrocatalytic measurements are presented in the Appendix A.

From the oxidation charge of the monolayer hydrogen, the electrochemically active Pt surface area (ECSA_Hupd_) can be calculated according to the Equation (1) [54]: ECSA_Hupd_ (cm^2^) = Q_oxHupd_ (µC)/210 (µC/cm^2^)(1)

The catalytic activity of the catalyst samples was tested in the ORR and the HOR by the rotating disc electrode (RDE) technique. A RDE is a glassy carbon working electrode used in a three-electrode system. The rotating speed of the electrode can be controlled, yielding a variable diffusion rate of the reactant. The measurements were done in 0.5 M H_2_SO_4_ solution (see details in the Appendix A). 

In the long-term stability test, the samples were submitted to cyclic polarization at a 100 mVs^−1^ scan rate for 10,000 cycles between 50 and 1000 mV potential limits.

The measure of the ECSA loss after 10,000 cycles of the stability test is the ΔECSA_10,000_ value defined in Equation (2) [50]:ΔECSA_10,000_ = {1 − (ECSA_10,000_/ECSA_1_)} × 100%(2)
where ECSA_10,000_ and ECSA_1_ are the values of the electrochemically active Pt surface area measured in the 10,000-cycle and the first cycle on the same sample (see Appendix A for details).

For comparison, activity in the HOR and ORR and the long-term stability of the commercial reference Pt/C electrocatalyst with 20 wt.% Pt loading were also studied by the same methods as described above.

## 3. Results and Discussion

### 3.1. Physicochemical Characterization of the Composite Supports and the Related Pt Electrocatalysts

Table 1 summarizes the results of the characterization of the Ti_0.8_Mo_0.2_O_2_-C composite supports and the related Pt catalysts by XRD and nitrogen adsorption measurements. Adsorption isotherms are shown in Appendix A. As can be seen in Table 1, S_BET_ values of the TiO_2_-based composite supports strongly depended on the type of the starting carbonaceous materials. The order of S_BET_ values of the composites is in line with the order of the specific surface area of the initial carbonaceous materials, however the S_BET_ values of the composites were significantly lower than those of the carbon backbones (1635 m^2^/g [55], 1344 m^2^/g [present work], and 245 m^2^/g [55] for BP, F-BP and V, respectively). The total pore volume values followed a similar trend.

The success of the synthesis of the Mo-doped composites with the appropriate structure was confirmed by XRD measurements, which provided information on the phase composition of the samples. According to the results of XRD measurements [50], the presence of the rutile phase was exclusive in the mixed oxide part of the composite (Table 1), which is favorable for the incorporation of the oxophilic metal Mo into the TiO_2_ lattice [43,45]. The characteristic distortion in the lattice parameters of the rutile phase that was obtained after HTT (*a* = 4.630 Å, *c =* 2.940 Å; pure rutile TiO_2_: *a* = 4.593 Å, *c =* 2.959 Å) confirms the incorporation of Mo into TiO_2_-rutile lattice with Mo_subst_ = 18% [50]. It has been suggested that the dissolution of Mo is prevented by its incorporation into the TiO_2_ lattice, which can contribute to the increased stability of the electrocatalysts during fuel cell operation [43,45,50]. It should be noted that no reflections characteristic to Mo oxides were found.

XRD patterns of the composite-supported electrocatalysts showed a broad band at 2 θ of 40 degree (Figure 1). We have demonstrated in our previous works that this band belongs to finely dispersed, uniformly distributed Pt particles with average size around 2–3 nm, due to the applied NaBH_4_–EG-assisted reduction–precipitation method [47,48,50]. In line with our previous observations, Pt loading by EG-assisted NaBH_4_ reduction did not affect the reflections of the TiO_2_-rutile phase [50]. Small peaks in the region of 2 θ of 23–25 degrees originated from the parent carbonaceous materials (see Appendix A).

TEM images of the selected mixed-oxide-containing composite-supported Pt electrocatalyst (Pt/75BP) show 2–4 nm quasi-spherical nanoparticles (NPs) that are evenly distributed in a BP carbon material (Figure 2A–D). Needle-like crystals are also present and tend to group in large flower-like formations (Figure 2A,E). The selected area diffraction (SAED) that was obtained on a representative area (Figure 2F) suggests that the needle-like crystals are mixed-oxide rutile crystals, while the small evenly distributed NPs indicated in the SAED pattern by a wide and diffuse ring, are metallic Pt. These aspects are confirmed both by XRD measurements and elemental mappings (see below on pages 9–10). The mixed oxide probably appeared in several morphologies, with smaller particles and larger flower-like crystallites; the Moiré effect arising from overlapping mixed oxide crystallites in the composite materials could also be observed (see Figure 2E). The onion-like structure of the BP carbon material could be well-recognized in Figure 2D. These observations were in accordance with our previous results [50], demonstrating good reproducibility of the preparation procedure.

The morphology of the Pt/75BP electrocatalyst was investigated by the SEM method (Figure 3). The thin film of the sample exhibited a granular structure. The different nanograins on the surface did not have large size variations and a porous structure between different grains was also evident.

The elemental composition of the Pt/75BP sample was evaluated by analyzing different sample regions by the EDX technique. EDX results obtained on the selected areas of the Pt/75BP electrocatalyst are shown in Appendix A. The composition data calculated from EDX and ICP-OES results are summarized in Table 2. Ti/Mo and mixed oxide/carbon (TiMoO_x_/C) ratios calculated both from the results of EDX and from ICP-OES measurements slightly differed from the nominal values. As shown in Table 2, there is some increase in the Ti/Mo ratios when the measured values are compared to nominal ones.

A decrease in the relative content of molybdenum has already been observed in our previous studies [43,50] and, apparently, can be a consequence of the partial dissolution of certain less stable Mo species (not incorporated into the TiO_2_ lattice) during the deposition of Pt. As can be seen from Table 2, the content of Pt in this electrocatalyst, measured by ICP-OES, was in line with the nominal value, but the Pt content obtained from the EDX data is highly dependent on the areas selected for inspection. In areas where the TiMoO_x_/C ratios are close to the nominal values, the Pt content appears to be significantly different from the nominal one. At the same time, the Pt content measured by the EDX method in the regions of the catalyst enriched in mixed oxide is in good agreement with the nominal value. This finding indicated that areas rich in Pt and poor in Pt may also exist on the surface of the composite supports and Pt nanoparticles have a high affinity to concentrate on the mixed oxide.

Elemental maps of the selected electrocatalyst (Pt/75BP) are presented in Figure 4. Patterns of Ti, Mo, and O were almost congruent (cf. Figure 4B–D), indicating the incorporation of Mo into the TiO_2_-rutile lattice. However, the Pt pattern was smeared indicating that the Pt settled on the carbonaceous part of the composite, too (cf. Figure 4E,F). Nevertheless, areas exist where Pt and the Mo doping elements are in close proximity to each other, providing the favorable interaction according to the bifunctional mechanism (see Figure 4G).

Figure 5 shows the Raman spectra of the as-prepared electrocatalysts used in this study. The composites present distinct Raman signatures for both Ti–O–Ti and C–C/C=C bonds. Thus, all spectra show the typical TiO_2_-rutile-type Raman-active optical phonon modes centered at 149, 420, and 608 cm^−1^, which are attributed to the B_1g_, E_g_, and A_1g_ modes, respectively. The band located at 254 cm^−1^, which is poorly developed in all our Raman spectra, represents a combined line typically appearing when the degree of distortion is high [56,57]. The weak high-frequency line of the B_2g_ symmetry, centered at 816 cm^−1^, was also poorly observed in our measurements. There are no lines corresponding to MoO_2_ present in the Raman spectra of our materials, suggesting a good and homogeneous incorporation of Mo within the structure.

On the other hand, the presence of graphitic C in the composites has been demonstrated by the corresponding D and G bands for the carbon materials, all spectra showing the first-order Raman lines at 1324 and 1592 cm^−1^ [58,59]. The D band corresponds to the disordered graphitic lattices that are usually assigned to K-point phonons of A_1g_ symmetry, while the G band is a signature of an ideal graphitic lattice, this former band being assigned to the Raman-active E_2g_ mode for the tangential in-plane stretching vibrations of the sp^2^-hybridized bond [60]. As all samples contain the same content (i.e., 75 wt.%) but from different types of carbon, it will be of interest to get information about the effect of their addition to the final composites. As is known, the I_D_/I_G_ ratio is a measure of the degree of defects present in the sample and the in-plane crystalline size of the sample [61]. Thus, the calculated I_D_/I_G_ ratios of Pt/75BP (1.24) and Pt/75V (1.20) are higher than that of the Pt/75F-BP (1.17), suggesting that the first sample has the highest degree of disorder; it possesses more defects and dislocations than the other two materials. It is important to observe that all samples show a high degree of disorder that normally is characterized by a broader G band as well as a broader D band of higher relative intensity when compared to that of the G band [59]. The in-plane crystallite size (L_a_) of the samples calculated from the Tuinstra–Koenig relation [61] (L_a_ (nm) = (2.4 × 10^−10^) λ^4^ (I_D_/I_G_)^−1^), where λ is the Raman excitation wavelength (633 nm in our case), were found to be 31, 33, and 32 nm for the Pt/75BP, Pt/75F-BP, and Pt/75V, respectively.

The surface composition of the electrocatalyst samples and the chemical state of the components was also investigated by XPS measurements. The apparent composition data (determined by assuming the homogeneous in-depth distribution of the components) are shown in Table 3. The data are generally consistent with the results of previous investigations [47,50]. The apparent carbon content in all electrocatalysts is somewhat higher than the nominal value. It is the result of the coexistence of the flower-like large oxide particles, which were noticed also in the TEM micrographs with more dispersed ones, leading to the inhomogeneous coverage of the carbon backbone. Nevertheless, the extent of this inhomogeneity is not large in the composites with 75 wt.% carbon, as the difference between the oxide/carbon ratio values that was determined by the surface specific XPS or the bulk-sensitive EDX/ICP-OES is small (see also Table 2). Another typical feature of the catalysts is that the apparent Pt content is well above the nominal value, while ICP-OES indicated a very similar Pt content to the nominal values (Table 2). This could be related to the well-dispersed nature of Pt on the outer surface of the supports, as confirmed by the XRD, TEM, and elemental distribution measurements. The Ti/Mo ratio measured by XPS is very close to both the data from bulk-sensitive measurements or the nominal value, which suggests the homogeneous incorporation of Mo into the mixed oxide, and is in agreement with the XRD data.

As expected, carbon is predominantly graphitic in all investigated electrocatalysts, characterized by a narrow but asymmetric C 1s line shape with a maximum at 284.4 eV binding energy. The Pt content is almost completely metallic with its 4f_7/2_ peak around 71.2–71.5 eV. The 458.8–459.0 eV binding energy of the Ti 2p_3/2_ peak indicates the exclusive presence of completely oxidized titanium in the mixed oxides. The complex Mo 3d line shape described in our previous works [46,62] was well reproduced in the studied electrocatalysts. The dominant contribution arises from Mo^6+^ ions (spin-orbit doublet with 3d_5/2_ binding energy at 232.5 eV [63]), but a pronounced asymmetry at the low binding energy side of the line shape indicates the presence of more reduced Mo species. In particular, a doublet with 3d_5/2_ binding energy around 231 eV arises from Mo^5+^ ions [63] or Mo^4+^ in a hydroxide- and/or molybdenum-bronze (Mo_x_(OH)_y_/H_4_MoO_4_)-like environment [64], while a feature with its lowest binding energy peak around 230 eV is assigned to Mo^4+^ [63,65] in the mixed oxide. In case of the Pt/75BP sample, a tiny metallic Mo contribution with its 3d_5/2_ peak slightly below 228.0 eV may also be present.

The fact that both the composition of the electrocatalysts and the chemical states of their components are very similar to those described in our earlier works demonstrates the highly reproducible nature of the catalyst synthesis procedure.

### 3.2. Electrochemical Characterization of the Pt/Ti_0.8_Mo_0.2_O_2_-C Electrocatalysts

In order to clarify the possibility of using the mixed oxide-containing 20 wt.%Pt/Ti_0.8_Mo_0.2_O_2_-C catalysts (C: V, BP and F-BP, Ti_0.8_Mo_0.2_O_2_/C: 25/75) as anode or cathode in PEM fuel cells, their electrochemical characteristics were investigated. The influence of the type of carbonaceous materials on the performance in the ORR and the HOR, expressed as current values normalized to the geometric surface area of the electrode, is compared in Figure 6. For comparison, the response of a commercial reference Pt/C catalyst is also shown under similar conditions.

Catalytic activity in the ORR after 10 cycles of the conditioning of Pt electrocatalysts was investigated by the RDE technique in O_2_-saturated 0.5 M H_2_SO_4_ solution (see Figure 6A). Potential dynamic polarization curves obtained by RDE measurements at six rotation speeds (225, 400, 625, 900, 1225, and 1600 rpm) for all catalysts are demonstrated in Appendix A. The curves in Appendix A show the expected increase in current densities at higher rotation rates indicating faster diffusion of oxygen onto the catalyst surface. At low potentials the current densities obviously depend on the rotating rates, indicating that the oxygen reduction is diffusion limited (Appendix A).

As shown in Figure 6A, the current density of the ORR in the mixed kinetic–diffusion controlled region, where the rate of the ORR reaction is limited by the availability of oxygen at the electrode surface, was higher on the fresh reference Pt/C catalyst, when compared to the composite-supported Pt catalysts. Identical diffusion limited currents were reached on the Pt/75F-BP and Pt/75V catalysts, whereas the limiting current of the Pt/75BP catalyst was slightly higher, which may be due to the different morphology of the support (e.g., as a result of aggregation of primary particles to secondary ones). Furthermore, the literature suggests that a smaller limiting current density can be originated from reversible oxide formation/reduction on Pt, which leads to the growth of the Pt particles and, consequently, to the reduction of the actual active surface area [66]. Marginally lower diffusion-limited current density in the composite-supported Pt catalysts can also be caused by a slower diffusion of oxygen through the oxide layer covering the Pt nanoparticles [40,67].

The onset potential for the ORR, i.e., the potential at which the reduction is started, was estimated for all catalysts. In accordance with the literature [68,69], the onset potential was identified by the change in the slope of the polarization curve due to transition from non-Faradaic to Faradaic activity. It should be noted that this method of determining the onset potential is not very precise, but it allows for a qualitative comparison of the activity of catalysts and distinguishes catalysts with high activity from less active catalysts. As shown in Figure 6A, very similar onset potentials for the ORR (~965 ± 10 mV) were observed for all catalysts, showing high activity in this reaction.

The kinetic parameters associated with ORR can be determined by means of a Koutecky–Levich (K–L) plot [70]. The limiting current density of diffusion, *j*_d_, is a function of the rotation rate (ω) of the electrode according to Equation (3) [71]:*j*_d_ = 0.2*nFD*^2/3^ν^−1/6^c_0_ω^1/2^ = Bω^1/2^(3)
where 0.2 applies when ω is expressed in rpm (1 rpm = 30/π radians/s), *n* is the number of electrons transferred in ORR per oxygen molecule, *F* is the Faraday constant (96,485 C/mol), *D* is the diffusion coefficient of oxygen in electrolyte (1.4 × 10^−5^ cm^2^/s), ν is the kinematic viscosity of sulfuric acid (1 × 10^−2^ cm^2^/s), and c_0_ is the bulk concentration of oxygen in the electrolyte (1.1 × 10^−6^ mol/cm^3^), which is assumed to be equal to its solubility (i.e., the solution is saturated).

The deconvolution of the measured current density, *j*, into the kinetic current density observable in the absence of any mass-transfer limit and diffusion current density, i.e., into *j*_k_ and *j*_d_, respectively, is accomplished with the Equation (4) [72,73]:1/*j* = 1/*j*_k_ + 1/*j*_d_ = 1/*j*_k_ + 1/(Bω^1/2^) (4)

The plot of 1/*j* vs. 1/ω^1/2^ should yield a straight line with an intercept equal to 1/*j*_k_, and 1/B represents the K–L slope (in the following discussion “1/B” will be identified as “K–L slope”). In agreement with the literature [74,75], the contribution of the current of diffusion through the Nafion film (*i*_f_) may be ignored, because the amount of Nafion used in this study was very small (ca. 20 µL of 5 wt.% Nafion in 5 mL of solution).

As shown in Figure 7, a linear relationship was obtained between the square root of the rotation rate (ω) and the current density, considering the *j* values in mA/cm^2^ measured at 300 mV potential; the linearity of the K–L plots confirms the reliability of the RDE measurements.

As emerges from Figure 7, the ORR proceeds with the same reaction mechanism on all catalysts studied. The slope of the plots is close to that observed for the theoretical four-electron transfer, even at such a high overpotential (300 mV vs. RHE), suggesting that the ORR proceeds via an overall mechanism leading to the direct formation of H_2_O. 

It has been mentioned in [76,77] that the interpretation of the RDE results obtained on electrocatalysts with a large surface area should be done with caution and the influence of the porous surface “architecture” upon the apparent electron–transfer kinetics of the ORR should be taken into account. Nevertheless, in the present study, catalysts were prepared and investigated using the same techniques, so that the RDE results can be interpreted by assuming the same surface architecture.

In contrast to the oxidation of hydrogen on Pt, which is a very fast reaction, the reduction of oxygen has sluggish kinetics and exhibits complex kinetic behavior. Tafel analysis is commonly used to compare electrocatalytic activity and to elucidate the reaction mechanism of electrocatalysts. In this method, the decimal logarithm of the current density plotted against the over potential (with respect to the thermodynamic one) is analyzed, which provides information related to the rate-determining step [78].

Conventionally, the Tafel analysis leads to two important physical parameters: the Tafel slope and the exchange current density [79,80]:η = *a* + *b* log(*j*)(5)
*a* = (2.3*RT*/α*nF*) log(*j*_0_)(6)
*b* = −(2.3*RT*/α*nF*)(7)
where the overpotential η represents the difference between the thermodynamic equilibrium potential and the applied potential (E − E_eq_), *a* contains the exchange current density (*j*_0_), and *b* is the Tafel slope expressed in mV/decade. The meaning of the other symbols is as follows: *n* is the stoichiometric number of electrons involved in an electrode reaction, α is the so-called transfer coefficient, *F* is the Faraday constant (charge on one mole of electrons) = 96,485 C/mol, *R* is the gas constant, equals to 8.314 J/(mol × K), while *T* is the absolute temperature = 298 K. The values of the exchange current density (*j*_0_) may be obtained by the extrapolation of the Tafel plot to the equilibrium potential for the O_2_ reduction (E_eq_ = 1.23 V).

It has been demonstrated that the Tafel slope in the ORR reaction changes with the applied potential [81]. As shown in Figure 8, the measured Tafel slope at a low overvoltage (region I) is ~60 mV/decade, which increases to ~120 mV/decade at a higher overvoltage (region II). In agreement with the literature [82,83], the former may reflect separate protonation of surface oxide species followed by its reduction, while the latter may be characteristic to the simultaneous protonation and electron transfer to surface oxides.

The results of the Tafel analysis performed for the electrocatalysts were presented in Appendix A. Despite the well-known fact that the Tafel plots are independent of the rotational speed in the range of 400–3600 rpm [84], the results obtained at rotational speeds of 900 and 1600 rpm were included in Appendix A for comparison. Since, at high overpotentials, diffusion effects begin to interfere with the kinetics of the electrode, in accordance with the recommendations given in reference [85], in the calculation overvoltage values not higher than η ≤ −413 mV were used.

The similarity between the Tafel slopes shown in Appendix A suggests the same reaction mechanism for the studied electrocatalysts. The best fit in the Tafel analysis in region II was obtained with a slope of −120 ± 1 mV/decade, supporting the transfer of one electron and simultaneous protonation in the rate-determining step according to Equation (8) [78], with the corresponding transfer coefficients equal to 0.49 and 0.50 for the composite-supported catalysts and Pt/C, respectively.
PtO_2_ + H_3_O^+^ + e^−^ → PtO_2_H + H_2_O(8)

While the Tafel slope provides insight into the reaction mechanism, the exchange current density is known as a descriptor of catalytic activity. The magnitude of the exchange current density determines how rapidly the electrochemical reaction can occur. As can be seen from Appendix A, at a Tafel slope of ~−120 mV/decade, the values of the exchange current densities, which depend not only on the Tafel slope, but also on the number of Pt sites involved, are in the range of 2.26 × 10^−4^ and 4.93 × 10^−4^ mA/cm^2^ and increase in the following order: Pt/C < Pt/75F-BP < Pt/75BP < Pt/75V.

Quite similar results can be found in the literature. Thus, in reference [86], the exchange current densities for the Pt/C and Pt–Fe/C alloy electrocatalysts with 60 wt.% Pt loading, determined at 60 °C in the region characterized by a −120 mV/decade Tafel slope, were 1.63 × 10^−5^ and 2.15 × 10^−4^ mA/cm^2^, respectively, thereby demonstrating the promoting effect of Fe in Pt–Fe bimetallic catalysts, resulting in an improved oxygen reduction activity.

According to the literature, the exchange current densities on various electrocatalysts in the regions I and II are ~10^−7^ and 10^−4^ mA/cm^2^, respectively. For instance, in reference [87] the *j*_0_ value obtained on the 40 wt.% Pt/C catalyst in the region I was three orders of magnitude lower when compared to the values obtained in the region II (5.25 × 10^−7^ and 3.16 × 10^−4^ mA/cm^2^, respectively). As emerges from Appendix A, with the exception of the Pt/75F-BP catalyst (*j*_0_= 4.11 × 10^−7^ mA/cm^2^), the exchange current density values on these catalysts in region I was within 1.08 × 10^−6^ and 1.84 × 10^−6^ mA/cm^2^. However, as can be seen from Figure 8, the region I was too short and therefore there may be a lot of uncertainty in the calculation.

Figure 6B displays HOR voltammograms (anodic scans) that were recorded via the RDE technique in H_2_-saturated 0.5 M H_2_SO_4_ (ν = 10 mV s^−1^, T = 25 °C and ω = 900 rpm) on the Pt/C and different carbonaceous materials—containing Pt/Ti_0.8_Mo_0.2_O_2_-C electrocatalysts. The currents were also normalized to the geometric area of the glassy carbon electrode.

It can be seen from Figure 6B that the electrochemical performance of the Pt/Ti_0.8_Mo_0.2_O_2_-C electrocatalysts was nearly identical, while the reference Pt/C catalyst shows a slightly lower HOR-limiting current. Potential dynamic polarization curves obtained on RDE at five rotation speeds (400, 625, 900, 1225, and 1600 rpm) for all catalysts are demonstrated in Appendix A.

It is well known that the rate of the oxidation of hydrogen on the Pt electrode in acid solution might be too fast to be measured with the RDE method [88,89,90]. For the polarization curves for HOR obtained in RDE measurements at room temperatures, the steady-state current generally rises sharply from the origin with a positive going potential and reaches a limiting plateau above 50 mV.

Considering the very high rate of HOR on platinum group metals and the low hydrogen solubility of the electrolyte, it is reasonable to assume that the measured currents even near to the equilibrium potential are mainly determined by the hydrogen diffusion rate of the solution. 

The K–L plot of *j*^−1^ versus ω^−1/2^ obtained from the results of the diffusion-limited potential region at 250 mV is shown in Figure 9. The intersection of the line at the ordinate is the reciprocal of the kinetic current density (1/*j*_k_). As can be seen from Figure 9, a rather similar slope 6.2 × 10^−2^ and 7.1 × 10^−2^ (mA/cm^2^) rpm^−1/2^ of the K–L plot was obtained for the V-containing (Pt/C and Pt/75V) and BP-containing (Pt/75BP and Pt/75F-BP) catalysts, respectively.

These values are close to the theoretical value 6.54 × 10^−2^ (mA/cm^2^) rpm^−1/2^ obtained in 0.5 M H_2_SO_4_ at 25 °C [91] and correspond to the experimental value for a smooth platinum electrode. Appendix A compares the literature data on the K–L slope values obtained on different catalysts in the HOR. Based on a comparison of these values, for example, the effectiveness of the various methods of catalyst deposition on a glassy carbon RDE can be evaluated (e.g., the results obtained on catalysts using the ideal smooth rotating disk electrode and the so-called thin-film RDE technique can be compared). Small differences in the calculation of the theoretical value of the K–L slope observed in Appendix A can be explained by the slightly different values used by various authors for the diffusion coefficient (*D*) and the bulk concentration of H_2_ in the electrolyte (c_0_).

Generally, the intercept of a dependence extrapolated to an infinite rotation rate with the ordinate corresponds to the reciprocal of the kinetic current density of the HOR at a potential of 250 mV. However, the uncertainty of such calculations is too high. It should be noted that the accuracy of estimating *j*_k_ values from the K–L plot will be better if an additional purification procedure (purification in oxidizing mixture, treatment in hydrogen, etc. [92]) is used. Moreover, it has been demonstrated [93] that a sufficiently precise evaluation of *j*_k_ can be done if the determined values are less than ca. 7 mA/cm^2^.

A typical cyclic voltammogram of Pt with the classical features of the adsorption/desorption of underpotentially deposited hydrogen between 50 and 350 mV along with a redox peak pair related to Mo between 380 and 530 mV was observed on all of the studied composite-supported Pt catalysts (see Figure 10A). According to the literature and our previous results [46,94,95] the appearance of these redox peaks in the voltammograms clearly confirm that there is an active interface between the Pt nanoparticles and the surface Mo species of the composite support. It should be noted, that, as shown in Figure 10A, the extent of the double layer capacity between 400 mV < E < 600 mV depends on the type of carbonaceous materials used for the preparation of the catalysts and correlates well with the BET surface area of the composite materials calculated on the basis of N_2_ adsorption measurements (see Table 1). 

The average values of the ECSA, obtained on fresh composite-supported catalysts from five different batches, which were calculated from the charge of the hydrogen desorption regions of the CVs, are presented in Table 4. A good reproducibility of the Pt loading obtained for the different batches of all the Pt/75C catalysts is demonstrated in Appendix A. The average ECSA value obtained on fresh reference Pt/C in four parallel measurements is also included in Table 4; the obtained results demonstrated good reproducibility of the measurements.

As can be seen from Table 4, the ECSA values calculated for BP-based electrocatalysts were slightly lower when compared to the catalysts prepared using V as the carbonaceous material (Pt/75V and Pt/C). The values of the ECSA loss after 10,000 cycles of the stability test (ΔECSA_10,000_) calculated from the charges originated from the hydrogen desorption in the 1st and 10,000th cycles are also included in Table 4. As shown in Table 4, the ΔECSA_10,000_ values confirm the higher stability of the BP-containing catalysts, which, as demonstrated in our recent study [50], can be explained by the more homogeneous microstructure of the catalysts with a high content of the high specific surface area carbonaceous starting material. However, as can be seen from Figure 10B, this difference in stability between these catalysts only begins to appear after 5000 cycles of the stability test.

Since the increased stability and CO-tolerance of this type of catalysts have already been demonstrated in detail in our recent study [50], we only want to emphasize the similarity of the ECSA values that were obtained after the 10,000-cycle stability test (compare the ECSA_10,000_ values in Table 4). Regardless of the electrochemically active Pt surface area values of the fresh samples, very similar values were obtained after the stability test with an average ECSA_10,000_ value of 50.2 ± 2.8 m^2^/g_Pt_ (R^2^ = 5.5%).

First, it is believed that larger Pt NPs are less prone to corrosion, and this observation is in good agreement with the assumption that the rate of degradation decreases with the age of the electrodes [96,97]. Thus, the similarity of ECSA values may mean that the Pt nanoparticles have reached their optimal size, which provides an increased stability to the catalyst. Moreover, these results confirm the assumption made in the ref. [98], demonstrating the fact that the stability of the catalyst under mild load cycles (cycling the potential between 0.6 and 1.0 V) mainly depends on the properties of the active Pt phase, while under start/stop cycle conditions between 1.0 and 1.5 V, the support properties determine the stability of the catalysts. Indeed, as demonstrated in our recent article [50,99], despite the decrease of the electrochemically active Pt surface area, the region of the electrochemical double-layer practically does not change, thus indicating that under mild load cycles (at potentials ≤ 1.0 V) carbon corrosion is not widespread.

It should be noted here that the comparison of long term stability data on different systems is often complicated by the application of different potential limits. In general, it is well known that oxophilic metal-containing catalysts become unstable at high potentials. Thus, in reference [100], a comparison of the electrocatalytic behavior of Pt/C, PtRu/C, and Pt/WO_x_ catalysts was carried out using different upper potential limits avoiding strong surface oxidation and irreversible damage to the oxophilic metal-containing electrocatalyst structure: 1.2 V was used as an upper potential limit for Pt/C and a lower potential (≤1.0 V) for Pt/WO_x_ and PtRu/C, which is sufficient considering the potentials that occur during normal operation of the PEM fuel cell (0.6–0.9 V). 

The high stability and corrosion resistance of Pt catalysts supported on rutile TiO_2_ when compared to Pt/C was also demonstrated by Dhanasekaran et al. [101]. It was found that serious degradation of the commercial 40 wt.% Pt/C occurs by cyclic polarization even upon using the upper potential limit of 1.0 V vs. RHE: after 10,000 cycles the loss in the ECSA was ca. 90 and 20% for Pt/C and Pt/TiO_2_ catalysts, respectively. It should be noted that despite the fact that the degree of degradation of the 40 wt.% Pt/TiO_2_ catalyst was very similar to the results obtained on our composite-supported catalysts (see Table 4), the active surface area of this catalyst was very low (18 m^2^ g^−1^).

## 4. Conclusions and Perspectives

Ti_0.8_Mo_0.2_O_2_–C composites with different types of carbon backbones (C: V, unmodified BP and functionalized F-BP; oxide/carbon: 25/75) were prepared by a sol-gel-based synthesis route for a detailed electrochemical study. The results of physicochemical and electrochemical characterization of the composite materials and the corresponding Pt catalysts demonstrated a good reproducibility of the synthesis procedure. 

The success of the synthesis of Mo-doped composites with appropriate structure was confirmed by XRD measurements and reinforced by Raman spectroscopy, indicating the complete incorporation of the Mo-dopant into substitutional sites of the rutile TiO_2_ lattice, preventing in this way the dissolution of Mo. TEM images confirmed the XRD and XPS results, showing the presence of well-dispersed and evenly distributed 2–4 nm Pt nanoparticles; the type of carbonaceous materials used in the synthesis of composite materials only slightly affected the average particle size of Pt. The Ti/Mo and mixed oxide/carbon ratios calculated from the results of EDX, XPS and ICP-OES measurements were consistent with the desired nominal values. Patterns of Ti, Mo, and O of the elemental maps were almost congruent, indicating the homogeneous incorporation of Mo into the rutile TiO_2_ lattice. 

Very similar onset potentials for the ORR (~965 ± 10 mV) were observed for commercial Pt/C and Mo-doped TiO_2_-containing Pt electrocatalysts, showing high activity in this reaction. The kinetic parameters related to ORR performance determined using the K–L plots suggested a four-electron transfer per oxygen molecule, leading to the formation of H_2_O. The similarity between the Tafel slopes suggested the same reaction mechanism for these electrocatalysts. The best fit in the Tafel analysis in region II was obtained with a slope of −120 ± 1 mV/decade, supporting the transfer of one electron and simultaneous protonation in the rate-determining step with the corresponding transfer coefficients being equal to 0.49 and 0.50 for the composite-supported catalysts and Pt/C, respectively.

The efficiency of the composite-supported electrocatalysts in the HOR was independent of the type of carbonaceous material used, while the reference Pt/C catalyst showed a slightly lower limiting current density. The slopes of the K–L plots were rather similar and close to the theoretical value: 6.2 × 10^−2^ and 7.1 × 10^−2^ (mA/cm^2^) rpm^−1/2^ for the V-containing (Pt/C and Pt/75V) and BP-containing (Pt/75BP and Pt/75F-BP) catalysts, respectively.

Among the catalysts studied, the BP-containing catalysts showed the highest stability. Regardless of the electrochemically active Pt surface area values of the fresh samples, under mild load cycles (at potentials ≤ 1.0 V) very similar values with an average ECSA_10,000_ value of 50.2 ± 2.8 m^2^/g_Pt_ (R^2^ = 5.5%) were obtained after the stability test on all studied electrocatalysts. According to these results, the Pt/75BP and/or the Pt/75F-BP catalysts can be recommended for further trials in fuel cell test experiments.

Our work delivers a solid base for further studies in exploring composite-supported catalysts as an alternative to more expensive materials and paves the way towards cleaner energy solutions. In particular, increased stability due to the composite support may allow researchers to reduce the amount of noble metals, thus contributing to the development of more affordable electrocatalysts for PEM fuel cells.

## Figures and Tables

**Figure 1 materials-15-03671-f001:**
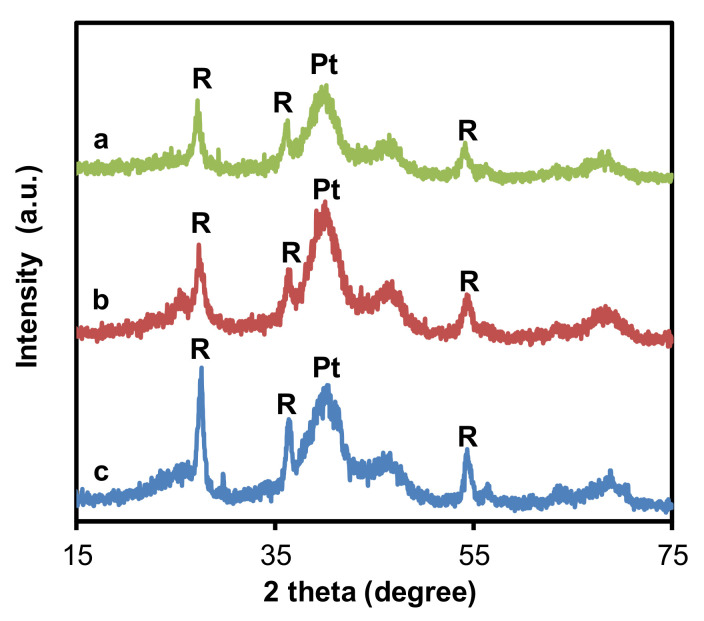
XRD patterns of the Pt/Ti_0.8_Mo_0.2_O_2_-C electrocatalysts with Ti_0.8_Mo_0.2_O_2_/C = 25/75 ratio: (a) Pt/75BP; (b) Pt/75F-BP; (c) Pt/75V; R: rutile, Pt: platinum.

**Figure 2 materials-15-03671-f002:**
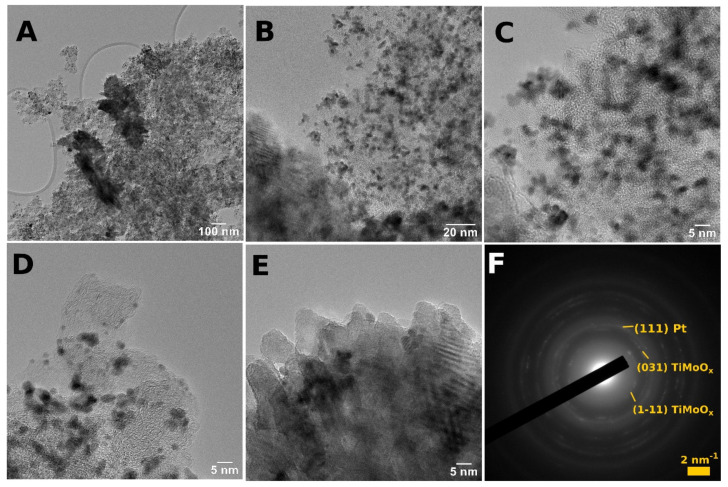
TEM images (**A**–**E**) and large-area SAED (**F**) of the Pt/75BP electrocatalyst.

**Figure 3 materials-15-03671-f003:**
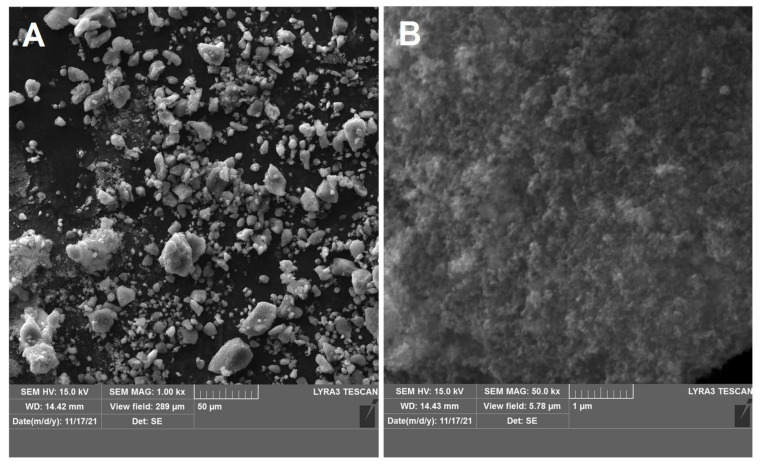
SEM images of the Pt/75BP electrocatalyst at lower and higher magnifications (**A**,**B**).

**Figure 4 materials-15-03671-f004:**
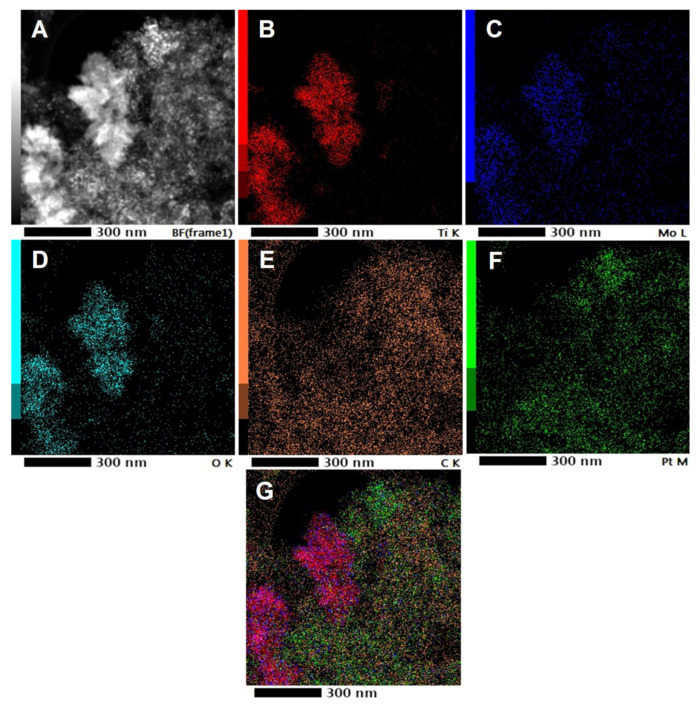
Elemental maps of the Pt/75BP electrocatalyst. HAADF STEM micrograph (**A**), Ti (**B**), Mo (**C**), O (**D**), C (**E**), Pt (**F**), and overview image (**G**).

**Figure 5 materials-15-03671-f005:**
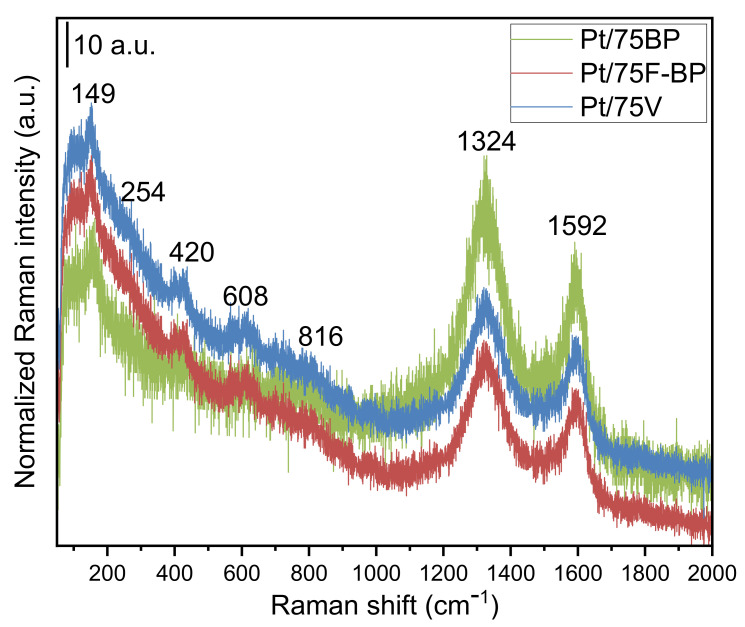
Raman spectra of the Pt/75BP, Pt/75F-BP and Pt/75V electrocatalysts.

**Figure 6 materials-15-03671-f006:**
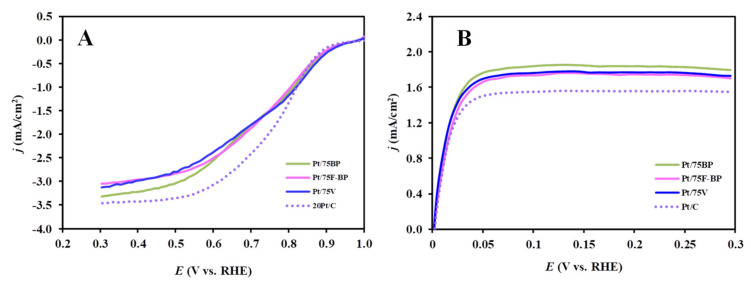
Electrochemical characterization of the reference Pt/C (dotted violet) and Mo-containing composite-supported Pt catalysts by RDE measurements at 900 rpm: Pt/75BP (green), Pt/75F-BP (pink), and Pt/75V (dark blue). (**A**) ORR curves obtained in O_2_-saturated 0.5 M H_2_SO_4_; (**B**) HOR curves obtained in a H_2_-saturated 0.5 M H_2_SO_4_. Obtained at 10 mV s^−1^, T = 25 °C.

**Figure 7 materials-15-03671-f007:**
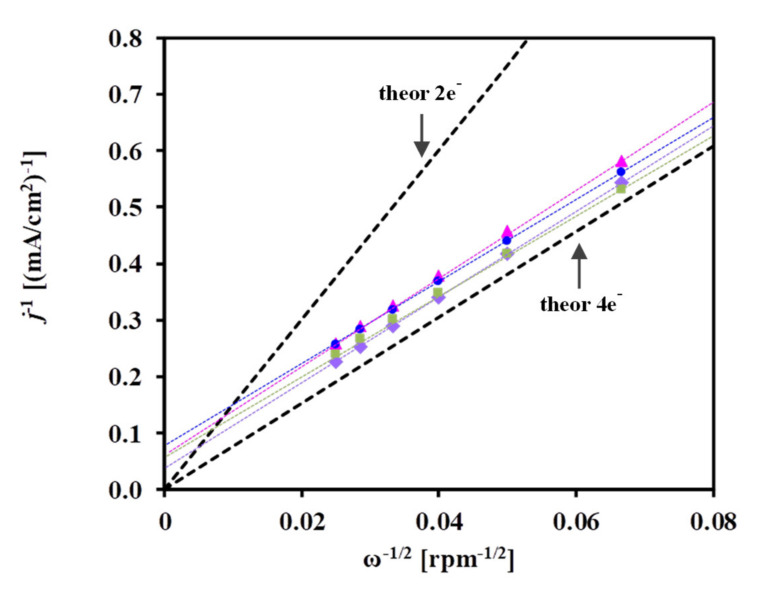
Koutecky–Levich representation of ORR measurements obtained at 300 mV vs. RHE on the (■) Pt/75BP, (▲) Pt/75F-BP, (●) Pt/75V, and (♦) Pt/C electrocatalysts.

**Figure 8 materials-15-03671-f008:**
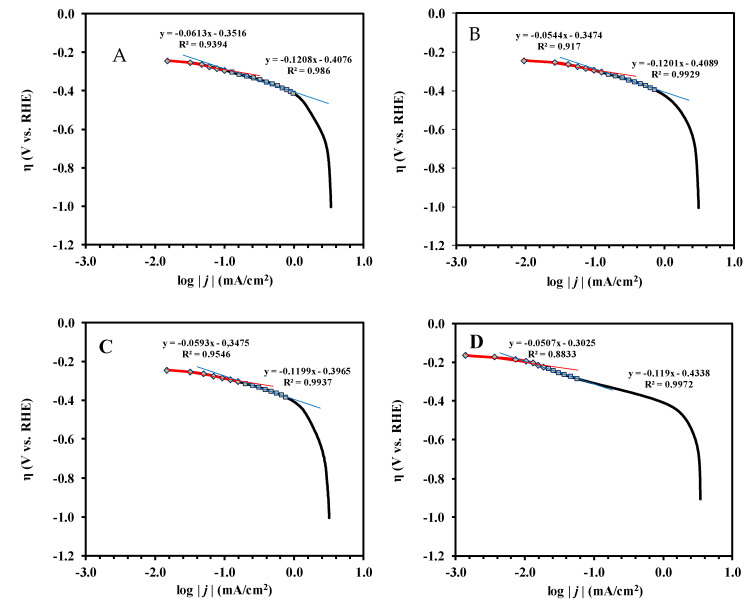
The Tafel plots for ORR measurements performed in 0.5 M H_2_SO_4_ at ω = 900 rpm on (**A**) Pt/75BP, (**B**) Pt/75F-BP, (**C**) Pt/75V, and (**D**) Pt/C electrocatalysts presented in Figure 6A; the straight lines correspond to different Tafel slopes.

**Figure 9 materials-15-03671-f009:**
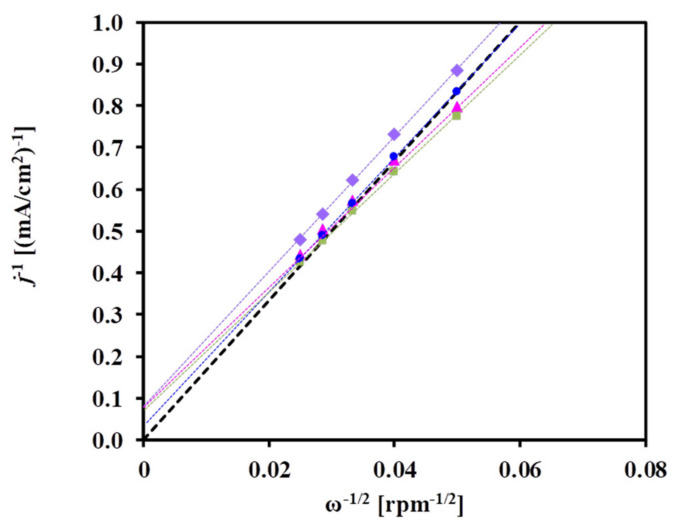
Koutecky–Levich representation of HOR measurements obtained at 250 mV on the (■) Pt/75BP, (▲) Pt/75F-BP, (●) Pt/75V, and (♦) Pt/C electrocatalysts.

**Figure 10 materials-15-03671-f010:**
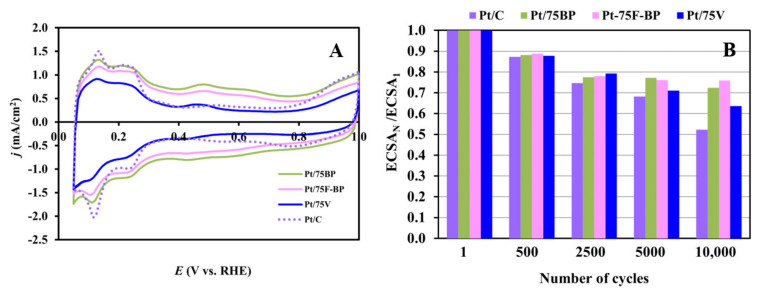
Cyclic voltammograms of the 20 wt.% Pt/75Ti_0.8_Mo_0.2_O_2_-25C electrocatalysts (**A**) and the results of the electrochemical long-term stability test; (**B**) comparison of the electrochemically active Pt surface area measured after N cycles, normalized to the ECSA measured in the 1st cycle (ECSA_N_/ECSA_1_) for the Pt/75BP (green), Pt/75F-BP (pink), and Pt/75V (dark blue) catalysts, as a function of the number of cycles (N); results obtained on the reference Pt/C catalyst (violet) are given for comparison. Recorded in 0.5 M H_2_SO_4_ at 100 mV·s^−1^, T = 25 °C.

**Table 1 materials-15-03671-t001:** Denomination, composition, and characterization of the composite supports and the related Pt catalysts by nitrogen adsorption and XRD measurements.

Sample ID ^(a)^	Nominal Composition of the Support	BET Surface Area, m^2^g^−1 (b)^	Pore Volume, cm^3^g^−1^	Rutile Lattice Parameters, Å ^(c)^	Pt Size, nm (XRD)
Pt/75BP	25 wt.% Ti_0.8_Mo_0.2_O_2_-75 wt.% BP	1120	2.01	*a* = 4.630, *c =* 2.940	2.68
Pt/75F-BP	25 wt.% Ti_0.8_Mo_0.2_O_2_-75 wt.% F-BP	726	1.32	*a* = 4.630, *c =* 2.940	2.75
Pt/75V	25 wt.% Ti_0.8_Mo_0.2_O_2_-75 wt.% V	175	0.48	*a* = 4.630, *c =* 2.940	2.08

^(a)^ BP: Black Pearls 2000, F-BP: functionalized BP carbon, V: Vulcan; ^(b)^ Specific surface area of the composite support materials determined by nitrogen adsorption measurements; ^(c)^ Lattice parameters of the rutile phase obtained after HTT; pure rutile TiO_2_: *a* = 4.593 Å, *c =* 2.959 Å.

**Table 2 materials-15-03671-t002:** Composition of the Pt/75BP electrocatalyst calculated from EDX and ICP-OES results in comparison with nominal values.

Method/Value	Ti/Mo (at/at) ^(a)^	TiMoO_x_/C (wt.%/wt.%)	Pt (wt.%)
Nominal	80/20	25/75	20.0
EDX ^(b)^	82.3/17.7	25.2/74.8	3.1
EDX ^(c)^	82.1/17.9	45.8/54.2	20.6
ICP-OES	83.8/16.2	18.7/81.3	19.2

^(a)^ Expected composition of Ti_0.8_Mo_0.2_O_2_ mixed oxide reflects the desired Ti/Mo atomic ratio; ^(b)^ Taken from sample region poor in Pt; ^(^^c)^ Taken from sample region rich in Pt.

**Table 3 materials-15-03671-t003:** Composition of the electrocatalyst samples calculated from XPS results in comparison with nominal values.

Sample ID ^(a)^	Ti/Mo (at/at)		Oxide/C (wt.%/wt.%)		Pt (wt.%)	
	Nominal	XPS	Nominal	XPS	Nominal	XPS
Pt/75BP	80/20	79.2/20.8	25/75	15/85	20	15
Pt/75F-BP	80/20	83.8/16.2	25/75	20/80	20	33
Pt/75V	80/20	80.5/19.5	25/75	19/81	20	42

^(a)^ Nominal composition of Ti_(1−x)_Mo_x_O_2_ mixed oxide reflects desired Ti/Mo atomic ratio.

**Table 4 materials-15-03671-t004:** Electrochemical performance of the reference Pt/C and Ti_0.8_Mo_0.2_O_2_-C composite-supported Pt catalysts.

Catalyst	ECSA_1_, ^(a)^ m^2^/g_Pt_	ECSA_10,000_, ^(b)^ m^2^/g_Pt_	ΔECSA_10,000_, ^(c)^ % ^(a)^
Pt/75BP	69.7 ± 2.6	50.1	27.6
Pt/75F-BP	70.9 ± 1.6	53.5	24.1
Pt/75V	78.3 ± 2.6	50.5	36.4
Pt/C	87.2 ± 2.3 ^(d)^	46.7	47.8

^(a)^ The average values of the ECSA obtained on fresh catalysts for five different batches; ^(b)^ The ECSA values obtained after 10,000-cycle stability test; ^(c)^ ΔECSA_10,000_ was calculated from the charges originated from the hydrogen desorption in the 1st and 10,000th cycles according to the Equation (2) (see Experimental part); ^(d)^ The average ECSA value obtained on fresh reference Pt/C in four parallel measurements.

## Data Availability

The data presented in this study are available on request from the corresponding author.

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
