# Peer review of "Electrocatalytic Properties of Mixed-Oxide-Containing Composite-Supported Platinum for Polymer Electrolyte Membrane (PEM) Fuel Cells"

_materials, 2022, doi:10.3390/ma15103671_

Round 1
Reviewer 1 Report
I reviewed the article. with title `` Electrocatalytic properties of mixed oxide containing composite supported platinum for polymer electrolyte membrane (PEM) fuel cells``. The theme of the article is very interesting and promising in the PEMs field but present article showing some serious issue which need to be corrected before publications. I am pleased to send you major level comments. The manuscript can be accepted for publication after modification. Please consider these comments/suggestions as listed below.
Comments:
- The title seems to be appropriate, but the idea appears to be jumbled. If possible, provide one introductory sentence describing your purpose at the beginning of your abstract.
- Introduction section must be written on more quality way, i.e., more up-to-date references addressed.
- In the beginning of introduction must explain about Polymer Electrolyte Membrane (PEM) fuel cells background.
- Research gap should be delivered on more clear way with directed necessity for the conducted research work
- The novelty of the work must be clearly addressed and discussed, compare your research with existing research findings and highlight novelty, (compare your work with existing research findings and highlight novelty.
- Please don’t use lumpy reference (such as [1-9]). Each reference needs to be properly addressed. Please revise your paper accordingly since same issue occurs on several spots in the paper.
- At the end of this sentence…… the start-up-shut-down driving of electric vehicles… cite these two reference to support your whole paragraph. (a) Modern trend of anodes in microbial fuel cells (MFCs): an overview (b) Electrode Material as Anode for Improving the Electrochemical Performance of Microbial Fuel Cells.
- ``The concept of non-noble metal-doped………. active carbon in a unique material system`` need a reference, cite these article as TiO2 already used in bioelectrochemical fuel cells. (a) Utilization of biomass-derived electrodes: a journey toward the high performance of microbial fuel cells (ii) Preparation, characterization, and application of modified carbonized lignin as an anode for sustainable microbial fuel cell.
- The main objective of the work must be written on the more clear and more concise way at the end of introduction section.
- Please write section 2.1 in paragraph form.
- Please stick to one abbreviation and be consistent with it in entire text. Please revise your paper accordingly since same issue occurs on several spots in the paper.
- Please extend the XRD and TEM explanation. There are several point to highlights which are missing in present article.
- Please provide high quality image for figure 1,2 and
- Why there are too many noise peak in raman? Please justify this error in your results.
- Regarding the replications, authors confirmed that replications of experiment were carried out. However, these results are not shown in the manuscript, how many replicated were carried out by experiment? Results seem to be related to a unique experiment. Please, clarify whether the results of this document are from a single experiment or from an average resulting from replications. If replicated were carried out, the use of average data is required as well as the standard deviation in the results and figures shown throughout the manuscript. In case of showing only one replicate explain why only one is shown and include the standard deviations.
- Please add a comparative profile section to compare your results and prove how it better than previous.
- Section 4 should be renamed by Conclusion and Future perspectives. Conclusion section is missing some perspective related to the future research work, quantify main research findings, highlight relevance of the work with respect to the field aspect.
- To avoid grammar and linguistic mistakes, Major level English language should be thoroughly checked. Please revise your paper accordingly since several language issue occurs on several spots in the paper.
- Reference formatting carefully need revision. All must be consistent in one formate. Please follow the journal guidelines to prepare the manuscript.
- The future perspective and present challenges should be discussing in separate section to make it more attractive to get more citations in future. If not, minimum author should include some future perspectives in conclusion section.
Author Response
The authors are grateful for the efforts of the Reviewer for enhancing the manuscript. Please find our detailed response to the questions and comments attached.

Reviewer 2 Report
In this manuscript, Ti1-xMoxO2-C composites are prepared by a sol-gel-based synthesis route for HOR and ORR applications. The catalytic properties are mainly determined by the interactions between Pt and Mo. The activity of the composite-supported electrocatalysts in the HOR was independent of the type of carbonaceous material used. Among the catalysts studied, the Black Pearls-containing catalysts showed the highest stability.
I suggest a minor revision for the manuscript since the content is suitable for the journal and the results are promising and interesting. The interpretation of some results should be more detailed and improved.
- For the Keywords, ‘sol-gel synthesis route’, ‘Tafel slope’, ‘reaction mechanism’, and ‘stability’ should be added to cause the interest of readers.
- Page 1, at the beginning of the Introduction, some basic concepts and introductions about PEMFC and its unique features should be introduced briefly. Since it is an electrochemical generator, what are the advantages of fuel cells over other techniques, such as lithium batteries and flow batteries [Energy Conversion and Management 207 (2020): 112514; Journal of Power Sources 493 (2021): 229445]? For example, how about the cost, energy density and safety issues compared with other techniques?
Moreover, what is the function of catalysts in PEMFC? Why is it so important, and why must use noble metals (such as Pt) as the electrocatalyst in PEMFC? This should also be briefly explained.
- Page 5, ‘All electrochemical measurements were carried out at ambient temperature.’It is suggested to directly indicate the ambient temperature in brackets.
- Page 6, ‘It has been suggested that the incorporation of Mo into the TiO2 lattice can prevent the dissolution of Mo and can contribute to the increased stability of the electrocatalysts during fuel cell operation, which plays an important role in the practical realization of these types of composite materials [15,17,22].’
There is something wrong with the logical expression of this sentence. The addition of Mo into the TiO2 can prevent the dissolution of Mo, but it seems the original system is without Mo, so where does the Mo dissolution come from? It should be ‘the incorporation of TiO2 lattice into Mo can prevent the dissolution of Mo’.
- Although the electrochemical characterization of this paper has been comprehensive, it is concluded that Black Pearls have the highest stability according to the electrochemical characterization results. However, in-situ fuel cell single-cell test experiments have not been carried out. Could the single-cell results be supplemented to further support the conclusions of the electrocatalytic tests?
Author Response
In general, the authors are grateful for the encouraging comments and recommendation of the Reviewer. Our response to the points raised is summarized in the attachment.

Reviewer 3 Report
The article presents an actual topic of increasing the stability of platinum-containing electrocatalysts based on a carbon support. The article needs to be finalized for the possibility of its publication.
- The synthesis of platinum electrocatalysts is carried out by the method of modified liquid-phase synthesis using borohydride as a reducing agent. It is known that the kinetics of the reduction reaction with borohydride is high even at room temperature, which leads to the production of large nanoparticles with an uneven distribution over the support surface. Why do the authors carry out reduction at elevated temperature?
- In Figure 1, for the Pt/75BP and Pt/75V samples, a small additional peak is observed near the first rutile peak (23-25 degree), which is not explained by the authors in the text.
- The caption of figure 6 shows what colors the different materials are marked with. There is a yellow color in the signature, while there is no potentiodynamic curve with such color in the figure. The same for Figure 10 - the colors of the CVs and histograms do not match the colors in the captions.
- Did the authors study the composition of materials after electrochemical measurements? (material from RDE).
- When discussing the activity in ORR (Figure 6b), the authors do not discuss the difference in the diffusion region between the commercial material and the resulting catalysts.
- Various electrochemical protocols are known to exist to assess stability. Stress testing in the potential range of 0.05 - 1.0 V is a catalyst protocol. In this range of potentials, degradation of platinum nanoparticles occurs predominantly. When it comes to increasing stability through the use of a composite carbon support, it is necessary to conduct stress testing in the potential range of 0.6 - 1.5 V or 0.6 - 1.4 V. This is considered to be the Support protocol.
- How did the type of CVs change before and after the stress test for all the studied materials.
- The work shows that the stability of the materials was evaluated by the change in the values ​​of ESA. It is important to study the final activity in the ORR and compare these values ​​with a commercial analogue.
Author Response
The authors are grateful for the efforts of the Reviewer for enhancing the manuscript. Please find our detailed response to the questions and comments in the attachment

Round 2
Reviewer 1 Report
The authors have addressed most of the reviewers' concerns in their revised manuscript. The referee supports its acceptance.
Reviewer 3 Report
I thank the authors for detailed responses to comments. The extensive comments of the authors and the changes made helped to significantly improve the quality of the presented study.
I recommend the article for publication in Materials